

# Source attribution of methane emissions from the Upper Silesian Coal Basin, Poland, using isotopic signatures

Alina Fiehn[1], Maximilian Eckl[1], Julian Kostinek[1], Michał Gałkowski[2,3], Christoph Gerbig[2], Michael Rothe[2], Thomas Röckmann[3], Malika Menoud[3], Hossein Maazallahi[3], Martina Schmidt[4], Piotr Korbeń[4], Jaroslaw Necki[5], Mila Stanisavljević[5,6], Justyna Swolkien[6], Andreas Fix[1], Anke Roiger[1]

[1] Deutsches Zentrum für Luft- und Raumfahrt (DLR), Institut für Physik der Atmosphäre, Oberpfaffenhofen, Germany
[2] Max Planck Institute for Biogeochemistry (MPI-BGC), Department of Biogeochemical Signals, Jena, Germany
[3] Institute for Marine and Atmospheric research Utrecht (IMAU), Utrecht University, Utrecht, The Netherlands
[4] Institute of Environmental Physics, University of Heidelberg, Heidelberg, Germany
[5] Faculty of Physics and Applied Computer Science, AGH University of Kraków, Kraków, Poland
[6] Faculty of Civil Engineering and Resource Management, AGH University of Kraków, Kraków, Poland

*Correspondence to*: Alina Fiehn (alina.fiehn@dlr.de)

**Abstract.** Anthropogenic emissions are the primary source of atmospheric methane ($CH_4$) growth. However, estimates of anthropogenic $CH_4$ emissions still show large uncertainties on global and regional scales. Differences in $CH_4$ isotopic source signatures $\delta^{13}C$ and $\delta^2H$ can help to constrain different source contributions (e.g. fossil, waste, agriculture, etc.). The Upper Silesian Coal Basin (USCB) represents one of the largest European $CH_4$ emission regions, with more than 500 Gg $CH_4$ yr$^{-1}$ released from more than 50 coal mine ventilation shafts and other anthropogenic sources. During the CoMet (Carbon Dioxide and Methane Mission) campaign in June 2018 methane observations were conducted from a variety of platforms including aircraft and cars to quantify these emissions. Beside the continuous sampling of atmospheric methane concentration, numerous air samples were taken from inside and around the ventilation shafts (1-2 km distance) and aboard the High Altitude and Long Range Research Aircraft (HALO) and DLR Cessna Caravan aircraft, and analyzed in the laboratory for the isotopic composition of $CH_4$.

The airborne samples downwind of the USCB contained methane from all sources in the region and thus enabled determining the mean signature of the USCB accurately. This mean isotopic signature of methane emissions was -50.9 ± 0.7 ‰ for $\delta^{13}C$



and -226 ± 9 ‰ for $\delta^2H$. This is in the range of previous USCB studies based on samples taken within the mines for $\delta^{13}C$, but more depleted in $\delta^2H$ than reported before. Signatures of methane enhancements sampled upwind of the mines and in the free troposphere clearly showed the influence of biogenic sources (e.g. wetlands, waste, ruminants). The ground-based samples taken during CoMet allowed determining the source signatures of individual coal mine ventilation shafts. These signatures displayed a considerable range between different shafts and also varied for individual shafts from day to day. Mean shaft signatures range from -60 ‰ to -42 ‰ for $\delta^{13}C$ and from -200 ‰ to -160 ‰ for $\delta^2H$. A gradient in the signatures of sub-regions of the USCB is reflected both in the aircraft data as well as in the ground samples with emissions from the southwest being most depleted in $\delta^2H$ and emissions from the south most depleted in $\delta^{13}C$. The average signature of -49.8 ± 5.7 ‰ in $\delta^{13}C$ and -184 ± 32 ‰ in $\delta^2H$ from the ventilation shafts fits with values from previous studies, but clearly differs from the USCB regional signature in $\delta^2H$. We assume that the USCB plume mainly contains fossil coal mine methane and biogenic methane from waste treatment, because the USCB is a highly industrialized region with few other possible methane sources. Assuming a biogenic methane signature between and -320 ‰ and -280 ‰ for $\delta^2H$, the biogenic methane emissions from the USCB account for 15-50 % of total emissions. The share of anthropogenic-biogenic emissions from this densely populated industrial region is underestimated in commonly used emission inventories. Generally, this study demonstrates the importance and usefulness of $\delta^2H$-$CH_4$ observations for methane source attribution, but highlights the need of comprehensive and extensive sampling from all possible source sectors.

## 1 Introduction

The 2015 Paris Agreement of the United Nations Framework Convention on Climate Change (UNFCCC) aims at limiting the rise in global mean temperature to 2°C. Additionally, the Global Methane Pledge has been signed by over 100 countries to reduce greenhouse gas emissions (European Commission and United States of America, 2021). To achieve this, we need to localize, quantify, and mitigate emissions of greenhouse gases due to anthropogenic activities (Ganesan et al., 2019; Nisbet et al., 2019; Nisbet et al., 2020). Methane ($CH_4$) is the second most important anthropogenic greenhouse gas after carbon dioxide ($CO_2$), and the increase in its atmospheric abundance since pre-industrial times has caused 23% of the radiative forcing of long-lived GHGs (Etminan et al., 2016). Reduction of methane emissions is attractive because of the relatively short lifetime of around 10 years, enabling relatively short-term results for mitigation policies (Dlugokencky et al., 2011; Nisbet et al., 2016). This has been recognized by policy makers and the European Commission has passed a regulation to reduce methane emissions from the energy sector, which puts the Union on a path to climate neutrality by 2050 (European Commission, 2021). A better understanding of methane emission sources helps to optimize potential mitigation pathways. While the total emissions can be constrained relatively well through top-down observations, there is still considerable uncertainty as to the contribution of individual source sectors (Saunois et al., 2020). Methane emissions can be of natural origin, like from wetlands in tropical and boreal areas, or from termites and wildfires. The anthropogenic sources include fossil fuel production and consumption, agriculture and waste management, biomass burning and biofuels.



The mean atmospheric CH$_4$ concentration has been rising since pre-industrial times with a short period of stagnation between 2000 and 2007 and an accelerated growth rate especially after 2014 (Dlugokencky et al., 2011; Nisbet et al., 2014; Nisbet et al., 2016; Nisbet et al., 2019; Saunois et al., 2020) and an even stronger surge since 2020 (Dlugokencky, 2022). This increase is caused by the imbalance of the methane sources and the tropospheric sinks, i.e. mainly the oxidation via its reaction with OH, but also to a much lesser extent transport to the stratosphere and uptake by soils. Which source or sink mainly causes the

observed increase is still under debate (Saunois et al., 2017; Nisbet et al., 2019; Lan et al., 2021).

The isotopic signatures of individual methane sources could help to understand the cause of the changes in emissions. The global mean ratio of the methane isotopologues in the atmosphere has been changing towards lighter carbon isotopic composition along with the rising concentration since 2007 (Nisbet et al., 2016). The ratio between $^{12}$C and $^{13}$C in the methane molecules and the ratio between $^2$H (= D, deuterium) and $^1$H atoms both differ for individual source categories of methane.

The atmospheric isotopic composition change is caused by changes in emissions from different sources. The debate on which sources caused the global increase in atmospheric concentration and decrease of $^{13}$C methane isotopes is still ongoing (Nisbet 2019). The isotopic information from different sources can be used in global inverse models to constrain the contribution of individual sources (e.g. Nisbet et al., 2016; Rice et al., 2016; Schwietzke et al., 2016; Rigby et al., 2017; Turner et al., 2017; Lan et al., 2021; Basu et al., 2022). To improve these model estimates, many studies collected and determined the isotopic

composition of various methane sources (Brownlow et al., 2017; Fisher et al., 2017; Sherwood et al., 2017; Menoud et al., 2021; Menoud et al., 2022b). The source signature observations were compiled into several databases to be readily available (Sherwood et al., 2017; Sherwood et al., 2020; Lan et al., 2021; Menoud et al., 2022a).

The isotopic composition accompanied by concentration observations can also be used to determine the relative strength of emissions from different sources in the same area. This has been done, for example, by Lu et al. (2021) for overlapping

emissions from gas production and cattle farming in Australia. Here we use the same approach for fossil coal mine and biogenic waste sector methane emissions; two sectors with great potential for emission mitigation. Waste sector emissions comprise about 18% of estimated global anthropogenic methane emissions of 366 Tg yr$^{-1}$ for the 2008-2017 decade (Saunois et al., 2020). This sector includes landfills and wastewater handling. In some countries the contribution of waste methane emissions to total anthropogenic emissions is much larger, i.e. in the U.S. 26% of anthropogenic emissions are from waste treatment

(USEPA, 2016). Coal mine methane emissions constitute about 42 (range 29-61) Tg yr$^{-1}$. This is a fraction of 11% of total global anthropogenic methane emissions for the 2008-2017 decade (Saunois et al., 2020). Most of the coal methane originates from underground hard coal exploitation. During mining methane is ventilated from the mines to keep the underground concentrations of methane below 2% to avoid explosions (Tchórzewski, 2017). Global emissions from coal mining are expected to keep increasing in the future because of the increasing mining depths and importance of abandoned coal mines

(Kholod et al., 2020). Poland is a country heavily depending on coal for its energy supply and industrial processes. Although it has reduced the fraction of energy from coal from 75% in 1990, still 40 % of energy were produced from coal in 2020 (International Energy Agency, 2022). This coal is predominantly mined in underground mines in the Upper Silesian Coal Basin (USCB) and also in the Lublin basin.



The isotopic composition of methane depends on the methane origin pathway (Whiticar, 1996). Thermogenic methane is
isotopically enriched ($\delta^{13}C$ > -50 ‰, $\delta_2H$ > -300 ‰) compared to biogenic methane ($\delta^{13}C$ < -50 ‰, $\delta_2H$ < -280 ‰), as
methanogens preferentially use the lightest isotopes due to the lower bond energy (Rice, 1993). The isotopic signatures of
methane from one coal mining area can vary significantly, which is connected with the fractionation of coalbed gases during
secondary, chemical and physical processes occurring during migration and/or mixing. The isotopic signatures of methane
from the USCB has been investigated in previous studies (Kotarba, 2001; Kotarba et al., 2002; Kotarba and Lewan, 2004;
Zazzeri et al., 2016; Menoud et al., 2021). The isotopic fractionation shows a difference between the northern and southern
part of the USCB and in the south also a depth relation, with isotopically lighter $\delta^{13}C$ methane at the top, which has resulted
from physical (e.g. diffusion and adsorption/desorption) processes during gas migration (Kotarba, 2001). Additionally, the
emission strength of the USCB coal mines has been thoroughly assessed with different methods during the CoMet 1.0
campaign in 2018 (e.g. Fiehn et al., 2020; Kostinek et al., 2021; Krautwurst et al., 2021; Andersen et al., 2023). During the
campaign, the isotopic signature of the well-mixed methane emissions from the USCB was determined from samples aboard
the German research aircraft HALO (Gałkowski et al., 2021b). Additionally, samples around the coal mine ventilation shafts
were taken and analyzed for isotopic methane composition in the framework of the Methane goes Mobile – Measurements and
Modeling (MEMO²) project. This project determined numerous isotopic source signatures of emission sources across Europe
with different techniques. The combined MEMO² data has been published in the European methane isotope database (EMID),
which includes all samples from the USCB coal mine ventilation methane (Menoud et al., 2022a).

In this study, we present isotopic methane sample analysis from a regional perspective of the USCB. We combine samples
from a smaller aircraft and ground samples to determine the contributions of coal mining and waste treatment to the total
USCB methane emissions. In Chapter 2 we present the observational data from airborne and ground-based sampling and the
method used to derive methane isotopic source signatures. Chapter 3 contains the results of the isotopic analysis for the airborne
samples, a comparison with ground samples and the source attribution to the source sectors. A summary and conclusions are
given in Chapter 4.

## 2   Data and methods

### 2.1   Airborne observational data

During the CoMet 1.0 campaign in early summer (May-June) 2018, several aircraft and ground-based instruments were
deployed to extensively investigate methane emissions of the USCB (Fix et al., 2018). Observations of methane dry air
concentrations and other trace species were conducted from several different airborne platforms, i.e. the German research
aircraft HALO (Gałkowski et al., 2021b), the DLR Cessna Caravan (Fiehn et al., 2020; Kostinek et al., 2021), and the Freie
Universität Berlin Cessna (Krautwurst et al., 2021). Additionally, the campaign was supported by observations of methane
concentrations from drones (Andersen et al., 2018; Andersen et al., 2023) and mobile in situ systems deployed in cars (Wietzel,
2018; Korbeń, 2021; Stanisavljević, 2021).



During CoMet 1.0, a total of ten flights were conducted with the DLR Cessna Caravan (Figure 1 and Table 1). Flight days were chosen according to the weather situation. Fair weather with as few clouds as possible and steady wind conditions were preferred to simplify mass balance analysis and to increase the temporal overlap with observations conducted with sunlight-dependent instruments. Depending on the wind direction, different parts of the USCB were targeted, with the objective to 135 determine emission estimates not only for the entire USCB, but also its parts. A focus region for sampling was the southwestern part of the USCB (Figure 2), since it contains some of the strongest emitting mines (e.g. Pniówek). The flights were designed as mass balance flights with an upwind track within the planetary boundary layer (PBL) and several legs downwind of the sources with the highest one just above the PBL. The optimal flight time for a mass balance is in the afternoon, when the PBL has reached its maximum extent and was vertically well-mixed. Four out of the ten flights were conducted on cloud-free 140 mornings in order to perform simultaneous observations with the MAMAP (Methane Airborne Mapper) instrument on the FUB Cessna. The airborne mass balance emission estimate for the entire USCB has been published in a previous study (Fiehn et al., 2020). Emission estimates of clusters of ventilation shafts were covered by the MAMAP instrument (Krautwurst et al., 2021). Using airborne in situ observations and dispersion modeling, Kostinek et al. (2021) were also able to estimate emissions of individual ventilation shafts during the CoMet 1.0 campaign. Andersen et al. (2023) determined the emissions of five 145 individual ventilation shafts and developed three upscaling methods to derive regional emission estimates. During the campaign period the wind direction varied considerably and all wind directions occurred. Flights were mostly conducted under easterly wind conditions.

Onboard the DLR Cessna Caravan a twin instrument to the Jena Air Sampler (JAS) from HALO (Gałkowski et al., 2021b) 150 was installed. It is an air sampler with drying unit and 12 glass flasks having a volume of 1 liter. Samples collected with both samplers were analyzed for trace gas concentrations ($CH_4$, $CO_2$, $CO$, $N_2O$, $H_2$, $SF_6$) and the isotopic composition of $CH_4$ and $CO_2$ ($\delta^{13}C$-$CO_2$, $\delta^{18}O$-$CO_2$, $\delta^{13}C$-$CH_4$, $\delta^{2}H$-$CH_4$) at the Max Planck Institute for Biogeochemistry (MPI-BGC) in Jena, Germany. Details of analyzed parameters and uncertainties are documented by Sperlich et al. (2016) and Gałkowski et al. (2021b). We report isotope ratios in the conventional $\delta$ notation as $\delta^{13}C = [^{13}R_{SA}/^{13}R_{ST} -1]$ and $\delta^{2}H = [^{2}R_{SA}/^{2}R_{ST} -1]$ where $^{13}R_i$ 155 and $^{2}R_i$ are the $^{13}C/^{12}C$ and $D/H$ ratios of a sample (i = SA) and an international standard (i = ST), respectively. The international standards are Vienna PeeDeeBelemnite (VPDB) for $\delta^{13}C$ measurements and Vienna Standard Mean Ocean Water (VSMOW) for $\delta^{2}H$ measurements. A total number of 62 flasks samples were successfully collected during nine flights in the USCB. We divided the samples according to the sampling location into three categories: free troposphere (FT), inflow (IN), and outflow/plumes (PL). PBL extent was estimated based on the location of the sharp vertical gradient of water vapor observed 160 in the in situ Cessna Caravan measurement data. Samples taken above the PBL are classified as free troposphere. Inflow and outflow samples were taken within the PBL and are classified either as inflow if they were taken upwind of the USCB coal mines or as outflow if they were sampled downwind of them. For each of these categories we determined the mean isotopic signature from all flights combined and for PL samples also for individual flights. In total, our dataset consists of 15 FT samples, 8 IN samples and 32 PL samples.







**Figure 1: Flight tracks of the ten CoMet 1.0 flights with the DLR Cessna Caravan in the USCB. Coal mine ventilation shafts from CoMet v4 emission dataset (Gałkowski et al., 2021a) and JAS sample locations are marked. © Google Earth**

## 2.2 Ground samples

On the ground the MEMO$^2$ teams sampled from several mobile platforms. Air samples from inside and around the mine ventilation shafts were taken in Supelco Flexfoil bags. The trace gas concentrations and isotopic composition of $CH_4$ was then analyzed by continuous flow isotopic ratio mass spectrometry at the Institute for Marine and Atmospheric Research Utrecht



(IMAU). The analysis is described in Röckmann et al. (2016). Measurements at IMAU and MPI-BGC are referenced to the JRAS-M16 reference gases (Sperlich et al., 2016).

Additionally, a Picarro G2201-i cavity ring-down spectrometer (CRDS) with an active air core system attached was used to determine δ¹³C from some CH₄ plumes observed by a measurement car (Wietzel, 2018; Hoheisel et al., 2019; Korbeń, 2021). Finally, the active air core samples from drones were also filled into sampling bags and analyzed for isotopic composition of CH₄ (Andersen et al., 2021). The data was synchronized and published in the EMID (Menoud et al., 2022b, a). More information on the sampling and measurement methods and all MEMO² isotopic signatures can be found therein.


**Figure 2: USCB map with ventilation shafts, waste installations including waste disposal and waste water treatment (CoMet v4 database) and locations of ground samples for individual shaft signature determination (Menoud et al., 2022a). The boxes mark the approximate target regions of different flights. © Google Earth**






### 2.3 Calculation of isotopic source signatures

The characteristic isotopic ratio of a specific methane source is also called the isotopic signature. The CH$_4$ from point sources mixes with the surrounding air in the atmosphere after it is released. The observed concentration of CH$_4$ around this source
$c_{obs}$ is a combination of the background concentration $c_{bg}$ and the concentration of the emissions from the source $c_s$.

$$\mathbf{c}_{obs} = \mathbf{c}_{bg} + \mathbf{c}_s \quad (1)$$

Likewise, the isotopic ratio of sampled CH$_4$ is a combination of the isotopic signatures of background and source CH$_4$, weighed with the respective concentrations.

$$\mathbf{c}_{obs}\,\boldsymbol{\delta}_{obs} = \mathbf{c}_{bg}\,\boldsymbol{\delta}_{bg} + \mathbf{c}_s\,\boldsymbol{\delta}_s \quad (2)$$

In the equation $\boldsymbol{\delta}_{obs}$, expressed using relative delta notation, is the observed isotopic signature, $\boldsymbol{\delta}_{bg}$ is the background signature, $\boldsymbol{\delta}_s$ is the emission source signature. The relationship between the isotopic ratio and the concentration of methane during the two-component dilution process is linear. To find the source isotopic signatures of methane emitters, the Keeling method makes use of this linear relationship (Equation 3).

$$\delta_{obs} = \boldsymbol{k}\,(\mathbf{1}/\boldsymbol{c}_{obs}) + \delta_s \text{ with } \boldsymbol{k} = \boldsymbol{c}_{bg}\big(\delta_{bg} - \delta_s\big) \quad (3)$$

$\boldsymbol{\delta}_{obs}$ and $\boldsymbol{c}_{obs}$ are analyzed from the samples and the source signature $\boldsymbol{\delta}_s$ is determined as the intercept of the linear regression (Keeling, 1958; Pataki et al., 2003). The slope $\boldsymbol{k}$ of the regression line contains the background characteristics, which need not be known for the Keeling method. The linear regression method chosen is the orthogonal distance regression (ODR), because it considers uncertainties in $\boldsymbol{\delta}_{obs}$ as well as in $\boldsymbol{c}_{obs}$. This method was used for all air samples taken in glass flasks in the aircraft. The isotopic signatures from ground samples collected within MEMO² presented here were derived using the same
methodology. Comparison of these estimates to ones obtained from a more robust Miller-Tans (Miller and Tans, 2003) method showed no significant differences (Menoud et al., 2022b).

## 3 Results and Discussion

### 3.1 Flight isotopic signatures

For the three categories (FT, IN, and PL) we determined the mean isotopic signature from all flights combined (Figure 3) and for PL samples also for individual flights (Figure 4).



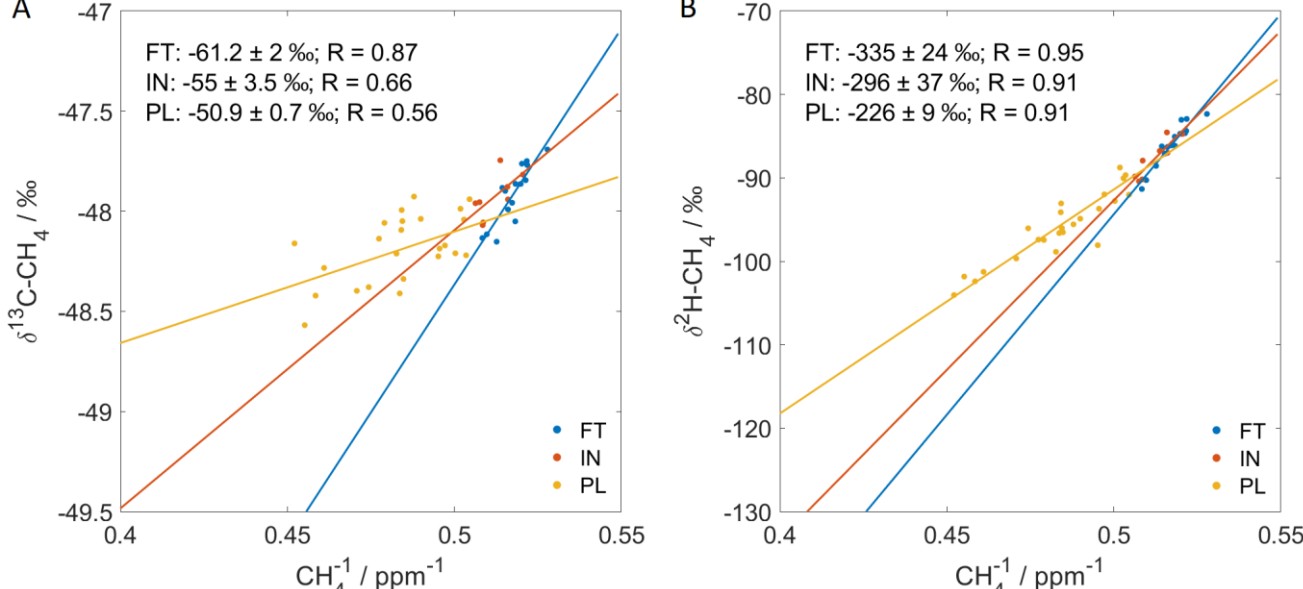

**Figure 3: Keeling plots for aircraft samples for δ¹³C (A) and δ²H (B) including source signature and Pearson correlation coefficients (R) for the three regimes free troposphere (FT), inflow (IN), and outflow/plumes (PL).**


The mean isotopic signature of the USCB is derived from all 32 samples collected inside the boundary layer downwind of the emission sources (Figure 3, PL). This average USCB signature is well constrained (-50.9 ± 0.7 ‰ δ¹³C and -226 ± 9 ‰ δ²H) with only small standard errors due to the large variance of samples contributing to the analysis. The samples from the inflow tracks and the free troposphere also showed a clear correlation between inverse methane concentration and isotopic ratios, and

the Keeling method could be applied for these samples as well, albeit with higher uncertainties. The observed $CH_4$ variability in the free troposphere originates from biogenic sources with a clear signature of -61.2 ± 2.0 ‰ δ¹³C and -335 ± 24 ‰ δ²H (Figure 4). In the free troposphere we encountered small variations in the $CH_4$ concentration from sources probably faraway, and most of them were biogenic (agriculture, waste, and wetlands). The signature of all inflow samples of -55.0 ± 3.5‰ δ¹³C and -296 ± 37 ‰ δ²H indicates that the $CH_4$ enhancements in the upwind boundary layer are mostly biogenic, but with a fossil

influence. In the PBL emissions upstream cause slightly larger $CH_4$ peaks that have more anthropogenic addition, as around Silesia there is industry and fossil fuel $CH_4$ emissions in all directions. The inflow samples might also be influenced by emissions from leaks in the natural gas networks in the area, which at that time also had a δ¹³C signature close to -55 ‰ (J. Necki, personal communication). The higher standard errors of this signature result from smaller concentration variations.

From the flasks taken within the boundary layer we also determined source signatures for individual flights. No samples were

collected in the study area during flights 4 and 7. During all other flights at least three flask samples within the PBL could be used to determine the source signatures using the Keeling method as described above. As mentioned previously, each flight



had a designated target region, which was either the entire USCB or a part of it. Table 1 lists all flights including wind direction, target region, number of samples, and isotopic source signature with standard errors. Figure 4 shows the location of the source signatures on a $\delta^{13}$C versus $\delta^2$H chart.

Flights 1 and 2 both covered the southwestern part of the USCB, where many deep and strongly emitting mines are located. These flights show the lowest $\delta^2$H signature of about -260 ‰. Flight 3 covered only the northern part of the USCB. Northerly winds provided a clean inflow. The wind was strong with a mean of 7 m s$^{-1}$, which caused a sampling of CH$_4$ plumes that were vertically not mixed from the ground to the PBL. The isotopic signature of flight 3 shows the highest value of -219 ‰ of all flights in $\delta^2$H. During flights 5 and 9 the conditions to sample the entire USCB were optimal and the plume was sampled with

sufficient distance to the sources of the plume to be well-mixed. The signatures of these two flights are very similar around -50 ‰ in $\delta^{13}$C and -230 ‰ in $\delta^2$H and probably represent the mean signature of the USCB CH$_4$ emissions. Flights 6 and 8 sampled emissions from the southern part of the USCB. These signatures are lighter in $\delta^{13}$C than those of the entire USCB. Flight 10 targeted two mines in the southeastern part of the USCB, called Brzeszcze and Silesia. The flight strategy followed a mass balance methodology, executed through circling around the mines. The four flask samples taken within the

PBL caught enhanced CH$_4$ from these two combined mines and allowed to determine their signature, albeit with a large uncertainty for $\delta^{13}$C. Overall, Figure 4 shows that signatures from the southern and southwestern regions have notably lower $\delta^{13}$C values. Also, the two flights covering the southwest of the USCB have reduced $\delta^2$H values. These gradients are compared to individual shaft signatures in the following.

**Table 1: Isotopic composition and standard error (SE) of CH$_4$ emissions for each flight alone and all flights combined.**

| Fl. | Date | WD | Target region | # Flasks in PBL | $\delta^{13}$C CH$_4$ [‰] | SE $\delta^{13}$C [‰] | $\delta^2$H CH$_4$ [‰] | SE $\delta^2$H [‰] |
|-----|------|----|----|----|----|----|----|----|
| 1 | 29.05.2018 | S | Southwest | 3 | -54.7 | 1.3 | -261 | 53 |
| 2 | 01.06.2018 | S | Southwest | 3 | -52.0 | 2.6 | -261 | 15 |
| 3 | 05.06.2018 | N | North | 7 | -49.4 | 1.3 | -219 | 12 |
| 4 | 06.06.2018 a | NE | Entire | 0 | - | - | - | - |
| 5 | 06.06.2018 b | NE | Entire | 7 | -49.6 | 2.0 | -228 | 24 |
| 6 | 07.06.2018 a | E | South | 3 | -52.4 | 1.8 | -223 | 12 |
| 7 | 07.06.2018 b | SE | Belchatow | 3 | - | - | - | - |
| 8 | 09.06.2018 | SE | South | 4 | -54.0 | 1.2 | -236 | 14 |
| 9 | 11.06.2018 | NW | Entire | 7 | -49.9 | 2.4 | -235 | 31 |
| 10 | 13.06.2018 | NE | Southeast | 4 | -49.3 | 7.6 | -237 | 10 |
| All | - | - | - | 32 | -50.9 | 0.7 | -226 | 9 |

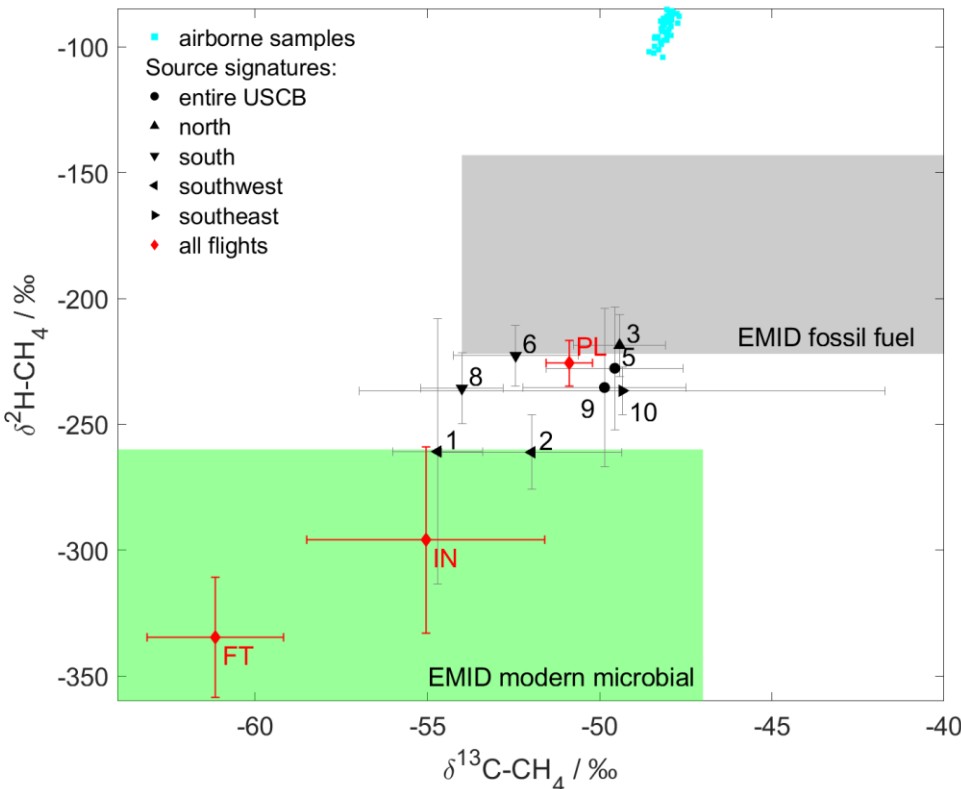

**Figure 4: Airborne samples and derived isotopic source signatures of CH₄ for each flight during the CoMet 1.0 campaign for the three regimes free troposphere (FT), inflow (IN), and outflow/plumes (PL). Numbers indicate the flight number, the symbol the target area. The colored areas indicate typical source signature ranges for fossil fuel (gray) and modern microbial (green) as the mean with one standard deviation from EMID (Menoud et al., 2022b).**

### 3.2 Comparison with ground isotopic signatures

The MEMO² team performed extensive CH₄ isotope sampling in the USCB in 2018 and 2019. Signatures were derived for individual sources within the USCB from samples in the vicinity and also from within the shafts. Biogenic emissions from a cow farm, two landfills, some manholes and a wastewater facility were also investigated. Although some biogenic samples were collected in Kraków, some 100 km to the east of main study area, we expect them to also be representative for similar types of sources in USCB. Coal mine methane signatures derived from samples taken on different days vary mostly within 50 ‰ for $\delta^2$H and up to 10 ‰ for $\delta^{13}$C (Figure 5). This variability may result from different areas of the mine being exploited as longwalls at different depths of the mine are opened up or shut down during excavation. At the Pniówek mine samples were taken inside the ventilation shafts in addition to the samples taken outside in the vicinity. $\delta^{13}$C signatures from all




samples are in the same range (not shown). Thus, the signature variability of the outside samples is reliable. For each shaft a mean signature is calculated from results on individual days. δ²H signatures of ventilation shafts are mostly within a range from -200 ‰ to -160 ‰. The mean δ¹³C values cover a range from -60 ‰ to -42 ‰, with one signature at -38 ‰.

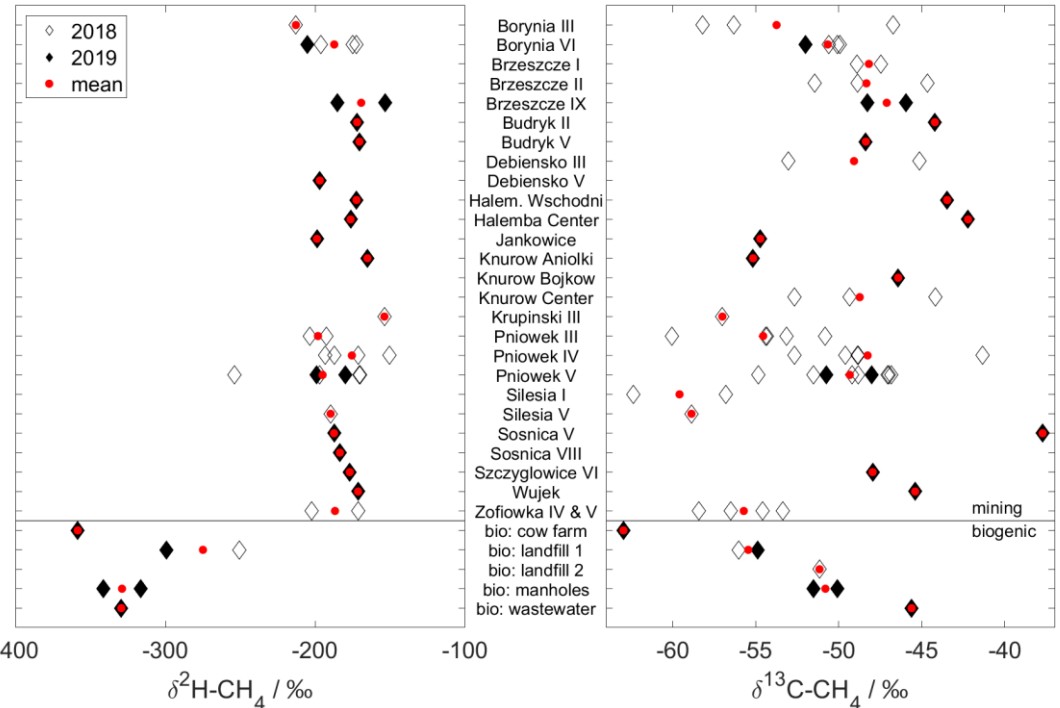


**Figure 5: Signatures of individual facilities from ground samples. Samples were taken on different days during two campaigns in 2018 and 2019. Diamonds show signatures on individual days and circles the mean signature from all days combined.**

Coal mine methane isotopic signatures are partly determined by coal attributes like deposition depth or physical parameters of the coal bed. With the comprehensive MEMO² dataset, we get a chance of investigating the variability within the USCB.

Looking at spatial gradients (Figure 6) a strong correlation (R = 0.66) is found for δ¹³C along latitude with southern mines' emissions being more depleted in δ¹³C. This tendency is also visible in the samples collected on the aircraft (Figure 4). There is no correlation detectable between δ¹³C and longitude in ground or aircraft samples. The correlation between δ²H of ground samples and latitude/longitude is moderate, and shows lowest signatures in the south and west. The aircraft samples showed average δ²H signatures for the southern region, but the southeastern mines had lower δ²H signatures than the entire USCB. In

summary, both ground-based individual shaft samples as well as the airborne sampling of subregions indicate emissions from the south being more depleted in δ¹³C and from the southwest being more depleted in δ²H.

The latitudinal δ¹³C gradient of the USCB is probably associated with its structural and lithostratigraphical history and generation and migration processes of coalbed gases, mainly methane (Kotarba, 2001; Kotarba and Lewan, 2004). The





indigenous coalbed gases in the USCB were generated during the Variscan thermogenic coalification process and subject to
intensive degassing to the surface in the following millions of years. In the central and northern parts of the USCB the
Mississippian and Pennsylvanian coalbed series are covered only by permeable strata and degassing continues to the present
day, explaining the low methane content of the coals in this region. The conditions in this region are not favorable for recent
generation of microbial methane and the thermogenic component of indigenous gases dominates. In the southern part of the
USCB, the coal-bearing strata were sealed with a clayey-sandstone cover during the Miocene. This practically impermeable
cover prevented the methane escaping to the surface and the gas accumulated below this layer causing the emissions in the
mines still to be high. This accumulation shows a lighter $\delta^{13}$C signature probably resulting both from additional gas created
through microbial $CO_2$-reduction processes and from fractionation of the indigenous gas during migration (diffusion and
adsorption/desorption) to the upper levels (Kotarba, 2001). This explains well the higher, more thermogenic, $\delta^{13}$C values in
the northern part of the USCB than in the south.

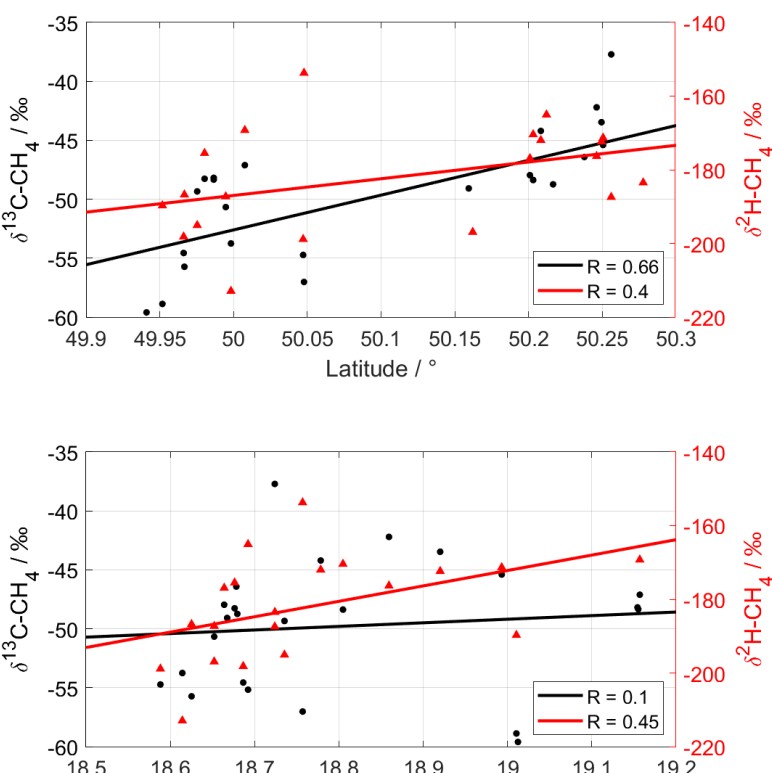


**Figure 6: Mean coal mine ventilation shaft signatures from the MEMO² dataset versus latitude and longitude to detect spatial gradients in the signatures within the USCB. In the legend the correlation coefficient is given.**



The coal mine methane emissions from the USCB have been isotopically characterized previously (Table 2).

The $\delta^{13}C$ signatures of individual shafts presented here are in the upper range and higher than previous signatures found by
(Kotarba, 2001; Kotarba and Lewan, 2004), while $\delta^{2}H$ signatures are in the range of previous signatures with some outliers with lower signatures. The EMID states an average isotopic signature for all active coal mines of -49.8 ± 5.7 ‰ in $\delta^{13}C$ and -184 ± 32 ‰ in $\delta^{2}H$ (Menoud et al., 2022a). This is well centered on the mean shaft signatures, but differs from the well mixed entire USCB methane plume observed on the Cessna aircraft and also the HALO aircraft (Gałkowski et al., 2021b). The two aircraft derived signatures match very well (Figure 7) and are shifted towards lower values with respect to the average coal
mining signature from the ground observations. This hints at an additional biogenic methane source within the USCB, that mixes with the coal mine methane and produces a different signature in the aircraft observations. This source will be evaluated in the next section.

**Table 2: Overview of literature values of USCB methane emission isotopic signatures with ranges or standard deviations.**

| Reference | $\delta^{13}C$-CH$_4$ [‰] | $\delta^{2}H$-CH$_4$ [‰] | Comment |
|---|---|---|---|
| Kotarba (2001) | -79.9 to -44.5 | -202 to -153 | Samples from boreholes inside the coal seam |
| Kotarba and Lewan (2004) | -72.8 to -47.8 | -196 to -153 | Samples from boreholes inside the coal seam |
| Zazzeri et al. (2016) | -50.9 ± 0.6 | | KWK Wujek deep mine shaft emissions |
| Gałkowski et al. (2021b) | -50.9 ± 1.1 | -224.7 ± 6.6 | CoMet 1.0 HALO aircraft observations, entire USCB (2 flights) |
| Stanisavljević (2021) | -50.2 ± 9.1 | -180.1 ± 38.3 | Weighted average of individual isotopic signatures (weighting by fluxes measured or reported by E-PRTR) |
| Menoud et al. (2022a) | -49.8 ± 5.7 | -184.0 ± 31.7 | Average of all active coal mine signatures in Silesia from the European data 2021 |




**Figure 7: Dual isotope plot for signatures of individual facilities together with USCB signatures derived from aircraft for the three regimes free troposphere (FT), inflow (IN), and outflow/plumes (PL) and other USCB literature signatures. Error bars denote standard deviations. The blue area shows the range of signatures from free gas inside the coal seam (Kotarba, 2001; Kotarba and Lewan, 2004). Shaded gray and green areas show the range of mean signatures with one standard deviation from for EMID fossil fuel and modern microbial methane sources, respectively (Menoud et al., 2022b).**

### 3.3 Emission attribution for the USCB

As depicted in Figure 7, the $\delta^2H$ signatures of ground samples and airborne samples for methane emissions from the USCB differ significantly. This means that the well-mixed plume sampled in the aircraft has another contributor with a different isotopic signature than the coal mine methane. Considering the location of the two signatures in Figure 7, this additional source is very likely a biogenic source with a potential contribution from natural gas leakage. Since the USCB is a heavily industrialized region with a sizeable population of around 3 million people, and agriculture only plays a minor role in this





region, most of the biogenic methane emissions probably originates from the waste sector (landfills and waste water
treatment). The share of this biogenic methane source in the USCB might be determined using the isotopic observations
from ground and aircraft using the same approach as in Lu et al. (2021).

Assuming that only biogenic and coal mine emissions contribute to the total methane emission of the USCB, the emissions
$F_i$ and the isotopic signatures $\delta_i$ fulfill Equations 4 and 5. From their combination follows Equation 6, which describes the
ratio of coal mining and biogenic emissions based on the isotopic signatures.

$$F_{coal} * \delta_{coal} + F_{bio} * \delta_{bio} = F_{tot} * \delta_{tot} \; ; \quad F_{coal} + F_{bio} = F_{tot} \qquad (4); (5)$$

$$\frac{F_{coal}}{F_{tot}} = \frac{\delta_{tot} - \delta_{bio}}{\delta_{coal} - \delta_{bio}} \qquad (6)$$

For our study we apply Equation 6 to the $\delta^2$H signatures, since these allow for higher discrimination than in the case of $\delta^{13}$C
(Figure 7). The observed signatures are $\delta^2$H$_{tot}$ = $\delta^2$H$_{aircraft}$ = -226 ± 9 ‰ and $\delta^2$H$_{coal}$ = -184 ± 32 ‰ (Menoud et al., 2022a).
The $\delta^2$H signature of biogenic emissions in the USCB is poorly constrained by measurements. The $\delta^2$H in methane emitted at
one cow farm (-358.7 ‰) is in the typical range of biogenic sources. The one landfill, for which $\delta^2$H observations (-275 ±
35 ‰) are available is not in the USCB directly, but located close to city of Kraków. There are no observations from waste
water treatment in the USCB, the listed manholes and waste water facility (both -329 ‰) in Figure 5 are also located in
Kraków. For comparison, the EMID includes $\delta^2$H signatures from 7 landfills (-275 ± 21 ‰) and signatures from 6
wastewater facilities (-323 ± 14 ‰) across Europe. The average signature over all these data points is -297 ± 30 ‰. The
mean $\delta^2$H value used for waste emissions in global modeling is around -300 ‰. Frank (2018) used a value of -304.3 ± 8.5 ‰
for landfill emissions considering signatures from several previous studies.

Considering these values, we assume that the USCB $\delta^2$H$_{bio}$ signature for waste emissions is -300 ± 20 ‰ for our study. The
total biogenic signature depends on the ratio of emission strengths between the landfills and the wastewater plants. Stronger
contribution from the landfills is suspected, but cannot be confirmed because of absent reporting or measurements, which
would shift the signature towards more positive values. Using Equation 6, the fraction of coal emissions is 50-85 % and of
biogenic emissions in the USCB is 15-50 % (Table 3). Stronger landfill emissions than wastewater plant emissions would
shift the ratio toward more coal emissions and less biogenic contribution. The emissions of methane in the USCB are mainly
caused by coal mining, but biogenic emissions seem to account for a non-negligible part, too.

**Table 3: Ratio of coal and biogenic emissions for different assumptions of the signature of biogenic emissions from the USCB.**

| $\delta^2$H$_{bio}$ | F$_{coal}$/F$_{tot}$ | F$_{bio}$/F$_{tot}$ |
|---|---|---|
| -280 | 85% | 15% |
| -300 | 62% | 38% |
| -320 | 50% | 50% |





The gridded emission inventories EDGAR v6.0 (Crippa et al., 2021) and CAMS-REG-GHG v3.1 (Granier et al., 2019) also
provide estimates per sector for anthropogenic emissions. According to them, in the USCB the methane emissions consist of
85% (CAMS) and 90% (EDGAR) emissions from fuel exploitation, mainly coal mining, with the remainder split between
the waste sector, agriculture and residential combustion (Figure 8). For CAMS the estimate of the share of biogenic

emissions is at the lower end of the result of our isotopic analysis or would be consistent with a signature of $\delta^2H_{bio}$
around -280 ‰. EDGAR clearly seems to underestimate biogenic emissions with only a 6% share. This underestimation has
also been noted for the Berlin metropolitan area (Klausner et al., 2020). Interestingly though, EDGAR does discriminate
between landfill and wastewater emissions and gives a ratio of roughly 1:1 for the USCB. The CoMet v4.01 emission
inventory (Gałkowski et al., 2021a) contains locations of 30 landfills and 24 waste water treatment plants in the USCB. Only

11 of these landfills listed emissions in the E-PRTR database (European Energy Agency, 2019) in the last years. Their
emission sum for 2018 is 2.8 kt $CH_4$ yr$^{-1}$, about 0.4% of the total USCB emissions from inventory data. The other 12
landfills were visually detected via Google Earth and are not listed in the E-PRTR. Landfills with cogeneration power units
do not report data to E-PRTR and assume that they emit less than the reporting threshold. Similarly, the 24 detected waste
water treatment plants in the USCB do not report to the E-PRTR either.

From the isotopic partitioning analysis, assuming a $\delta^2H_{bio}$ signature between -320 ‰ and -280 ‰, anthropogenic biogenic
emission in the USCB is 15-50 % of $CH_4$ emissions and, thus, seem to be underestimated in heavily populated industrial
regions in gridded and point source emission inventories.

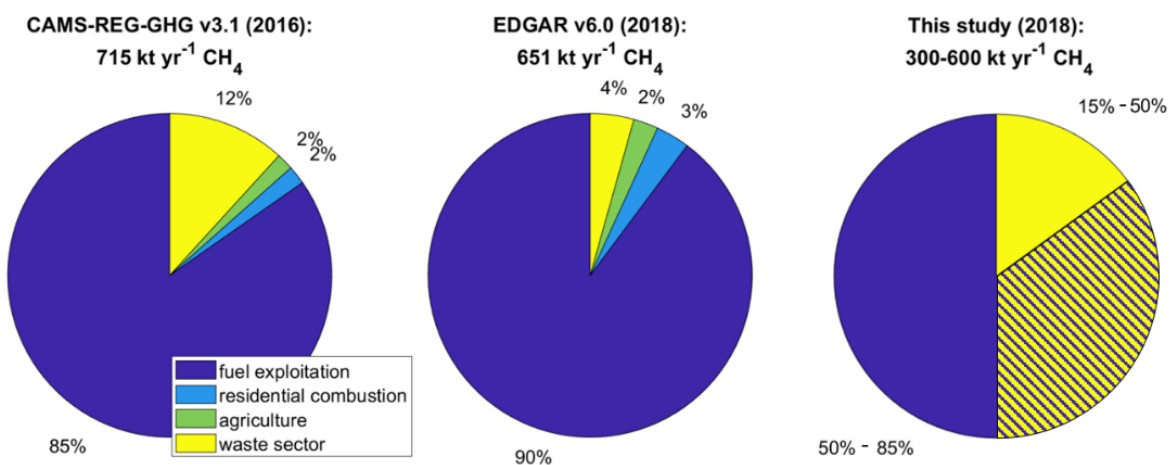

**Figure 8: Sectorial partitioning of methane emissions from the USCB according to CAMS-REG-GHG v3.1 (2016) and EDGAR
v6.0 (2018) emission inventories. Emissions were averaged over an area from 18.0°-19.6°E and 49.6°-50.5°N. The total emissions
for our study are derived via airborne mass-balance during the CoMet 1.0 campaign (Fiehn et al., 2020). The yellow-blue shaded
area indicates the uncertainty of the present data.**



## 4    Summary and Conclusions

In times of rising atmospheric concentrations of greenhouse gases, and countries trying to reduce their associated emissions, it is important to locate, quantify, and mitigate emissions of greenhouse gases due to anthropogenic activities. Differences in $CH_4$ isotopic source signatures $\delta^{13}C$ and $\delta^2H$ can help to constrain different source contributions (e.g. fossil, thermogenic, or biogenic). In the Upper Silesian Coal Basin, which represents one of the largest European $CH_4$ source regions, methane is emitted from more than 50 ventilation shafts of the underground mines. But as a heavily industrialized area with more than 3 million inhabitants there is probably also considerable contribution from the waste sector.

During the CoMet (Carbon Dioxide and Methane Mission) campaign in June 2018 methane observations were conducted from a variety of platforms including aircraft and cars. Beside the continuous sampling of atmospheric methane concentration, numerous spot air samples were taken from inside the ventilation shafts, in their immediate vicinity (1-2 km distance) and aboard the DLR Cessna Caravan aircraft, and analyzed in the laboratory for the isotopic composition of $CH_4$. Isotopic source signatures of $\delta^{13}C$ and $\delta^2H$ were determined using the Keeling method.

The airborne samples were divided into three categories according to the sampling location: free troposphere, and inflow and outflow/plumes within the boundary layer. Mean isotopic source signatures were determined for all three categories. The free troposphere methane originates from biogenic sources with a clear signature of -61.2 ± 2.0 ‰ $\delta^{13}C$ and -335 ± 24 ‰ $\delta^2H$. The signature of all inflow samples of -55.0 ± 3.5‰ $\delta^{13}C$ and -296 ± 37 ‰ $\delta^2H$ shows that the methane enhancements in the upwind boundary layer are mostly biogenic, but with an additional fossil influence. Due to prevailing easterly winds during the campaign, this result applies to sources to the east of the USCB. Samples collected in the boundary layer from a well-mixed plume downwind of the USCB allowed for the accurate determination of the signature of the entire USCB region, equal to -50.9 ± 0.7 ‰ $\delta^{13}C$ and -226 ± 9 ‰ $\delta^2H$. This is in between the range of typical microbial and thermogenic coal reservoirs, but more depleted in $\delta^2H$ than previous USCB studies reported based on samples taken within the mines. Source signatures could also be determined for the individual flights of the campaign, which focused on emissions from individual sub-regions. The ground-based samples allowed determining the source signatures of individual ventilation shafts. Their signatures vary strongly from mine to mine and even shaft to shaft and over time. A regional gradient in the signatures of sub-regions of the USCB is reflected both in the aircraft data as well as in the ground samples with emissions from the southwest being most depleted in $\delta^{13}C$ and $\delta^2H$. This gradient reflects the geographical structure of the USCB and the generation and migration processes of $CH_4$. The average signature from the ventilation shafts of -49.8 ± 5.7 ‰ in $\delta^{13}C$ and -184 ± 32 ‰ in $\delta^2H$ clearly differs from the total region signature in the $\delta^2H$, but fits well with values from previous studies. We assume that the regional plume mainly contains coal mine methane and biogenic methane from waste treatment and a $\delta^2H_{bio}$ signature between -320 ‰ and -280 ‰. Emissions from agriculture were considered negligible and excluded from the estimate. The differences in $\delta^2H$ signatures from airborne and ground-based averages then imply that the emissions of methane in the USCB are mainly caused by coal mining, but biogenic emissions seem to account for a significant part of 15-50% as well. The large uncertainty range of this result is caused by the uncertainty of the exact isotopic signature of the biogenic source, which in turn results from the small number of biogenic



samples and the uncertainty of emissions distribution between landfills and wastewater treatment facilities. The contribution of biogenic methane is underestimated in the point source and gridded emissions inventories E-PRTR, CAMS-REG and

EDGAR, which give biogenic fractions of 0.4-14% for this region. The inventories seem to generally underestimate emissions from the waste sector in heavily populated industrial regions.

This study confirms the importance and usefulness of $\delta^2$H-CH$_4$ observations for methane source attribution. These results should be corroborated with more observations of $\delta^2$H$_{bio}$ signatures in the USCB and other population centers.

**Author contributions:** Alina Fiehn, Maximilian Eckl and Julian Kostinek collected the air samples on the Cessna Caravan. Michał Gałkowski and Christoph Gerbig measured and analyzed the airborne samples. Alina Fiehn interpreted the data and wrote900 the manuscript. Thomas Röckmann coordinated the deployment of ground-based measurements and helped with data evaluation and interpretation. Malika Menoud, Hossein Maazallahi, Martina Schmidt, Piotr Korben, Jaroslaw Necki, Mila Stanisavljevic, and Justyna Swolkien conducted ground-based in situ measurements in the field during the campaign and

collected and shared the data. Andreas Fix coordinated all CoMet campaign contributions. Anke Roiger developed the research idea and coordinated the CoMet Cessna campaign operations. All authors contributed to the interpretation of the results and the improvement of the manuscript.

**Competing interests:** At least one of the (co-)authors is a member of the editorial board of Atmospheric Chemistry and

Physics.

**Acknowledgement:** The authors especially thank DLR-FX for the campaign cooperation, especially the pilots Thomas van Marwick and Philipp Weber and the group of Ralph Helmes, Andreas Giez, Martin Zöger, and Martin Sedlmeir. We thank DLR VO-R for funding the young investigator research group "Greenhouse Gases". We acknowledge funding for the CoMet

campaign by BMBF (German Federal Ministry of Education and Research) through AIRSPACE (FKZ grants no. 01LK1701A and 01LK1701C). The ground-based measurements on vehicles were funded by the European Union's Horizon 2020 research and innovation program under the Marie Skłodowska-Curie ITN project Methane goes Mobile – Measurements and Modelling (MEMO2; https://h2020-memo2.eu/) grant agreement no. 722479. Part of the research results presented in this paper have been developed with the use of equipment also financed from the funds of the "Excellence Initiative - Research University"

program at AGH University of Krakow. We would also like to kindly thank all members of IsoLab and Gaslab at MPI-BGC in Jena for their work in analyses of CoMet airborne samples, especially Armin Jordan and Heiko Moossen. We thank Maciej Kotarba for valuable help with the discussion of USCB geography and coalbed gases.

**Data Accessibility:** The data will be available from open access databases.



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
