# Peer review of "Source attribution of methane emissions from the Upper Silesian Coal Basin, Poland, using isotopic signatures"

_EGUsphere, 2023_

## Referee Comment (RC1)

The manuscript "Source attribution of methane emissions from the Upper Silesian Coal Basin, Poland, using isotopic signatures" by Alina Fiehn et al., calculates the isotopic source signatures of $\delta^{13}C$ and $\delta^2H$ from the airborne samples and the ground-based samples using the Keeling method which later helps to constrain the different source contributions for coal and biogenic emissions.

Overall, this work provides important information and data about isotopic source signatures. The isotopic signatures analysis is presented in a straightforward way which is easy to understand. The manuscript is written and structured well. Therefore, I suggest it to be suitable for publication in ACP after addressing specific and technical comments as listed below.

**Specific comment:**

1. L210: how to obtain the uncertainty of $\delta^{13}C$ showed in Figure 3? $\delta^{13}C$ is the interception for the fitting line and how do the authors define the uncertainty?

2. L224: authors mentioned that "the signature of all inflow samples....indicates that the $CH_4$ enhancements in the upwind boundary layer are mostly biogenic, but with a fossil influence". But from figure 4, the IN symbol and its error bars are all located in green shaded area, i.e., EMID modern microbial. I do not fully understand why there is a fossil influence.

3. L 250, Table 1: are the values of the isotopic composition and standard error originated from all flasks during each flight or only from the flasks in PBL? Please specify it.
I suppose the symbol "#" presents the number of flasks. Maybe No. is better for understanding.

4. L253, Figure 4: I think the individual source signatures (with black symbols and numbers) represent the results originate from the flasks in PBL. If so, please specify this information, otherwise, it is misleading.

5. L267: "$\delta^{13}C$ signatures from all samples are in the same range (not shown)". It would be interesting to see the results. Maybe put it in the appendix.

6. L269: "...with one signature at -38 ‰." I assume that you mean this value is out of the main range. Perhaps change it to "with one biased signature at -38 ‰".

7. L278: "The aircraft samples showed average $\delta^2H$ signatures for the southern region, but the southeastern mines had lower $\delta^2H$ signatures than the entire USCB." I assume the aircraft samples indicate the dots in Figure 3, and the cyan square symbols in Figure 4. If so, in my opinion it would be beneficial to show another figure (like in the appendix) to better present the statement. The coordinate-related information cannot be found from Figure 3 or 4. If the "aircraft samples" represent the isotopic source signatures in PBL shown in Figure 4., it is the southwest but not the southeastern mines having lower $\delta^2H$ signatures. Please comment.

8. L339: "For comparison, the EMID includes $\delta^2H$ signatures from 7 landfills (-275 ± 21 ‰) and signatures from 6 wastewater facilities (-323 ± 14 ‰) across Europe. The average signature over all these data points is -297 ± 30 ‰." I assume that the average signature from the 7+6=13 sites is calculated based on weighted average. How about the uncertainty? Is the error propagation or other method used? Please comment.

9. L343: "we assume that the USCB $\delta^2H$bio signature for waste emissions is -300 ± 20 ‰ for our study". Is the mean value of -300‰ from the global modeling as mentioned in L341? This value does not match with the mean value of biogenic signatures shown in Figure 7. Additionally, from where the uncertainty of 20‰ come? Please specify.

10. L351: CAMS-REG-GHG inventory has been updated to v5.3 in 2022. There might be no huge difference between v3.1 and v5.3. It is for your information.
    It would be also beneficial to compare the spatial distribution of gridded inventory and the results here. Will the CAMS inventory in the southwest area tends to have more biogenic sources?

**Technical comment:**

11. L100: the subscript in the "$\delta_2H$" should be superscript.

12. L162: not fully understand the sentence "For each of these categories we determined the mean isotopic signature from all flights combined and for PL samples also for individual flights." Please rephrase.

13. L199: please keep consistent format for $\delta_x$ in equation(3) and afterwards. The $\delta_{obs}$, $\delta_{bg}$ are in bold in previous text.

14. L264: I think this sentence has a grammatical error. "This variability may result from different areas of the mine being exploited as longwalls at different depths of the mine are opened up or shut down during excavation."
Maybe change to "This variability may result from different areas of the mine during longwall exploitation at different depths of the mine which are opened up or shut down during excavation" Please rephrase.

---

## Author Comment (AC1)

**Answer to reviewer #1 for "Source attribution of methane emissions from the Upper Silesian Coal Basin, Poland, using isotopic signatures"**

*We would like to thank the reviewer for the suggestions to improve the manuscript. Below you find our answers to their comments. The reviewer's comments are written in normal font, our answers in italics.*

The manuscript "Source attribution of methane emissions from the Upper Silesian Coal Basin, Poland, using isotopic signatures" by Alina Fiehn et al., calculates the isotopic source signatures of $\delta^{13}C$ and $\delta^{2}H$ from the airborne samples and the ground-based samples using the Keeling method which later helps to constrain the different source contributions for coal and biogenic emissions.

Overall, this work provides important information and data about isotopic source signatures. The isotopic signatures analysis is presented in a straightforward way which is easy to understand. The manuscript is written and structured well. Therefore, I suggest it to be suitable for publication in ACP after addressing specific and technical comments as listed below.

*We thank the reviewer for this positive statement.*

**Specific comment:**
1. L210: how to obtain the uncertainty of $\delta_{13}C$ showed in Figure 3? $\delta_{13}C$ is the interception for the fitting line and how do the authors define the uncertainty?
*The standard errors of the intercepts or $\delta_{13}C$signatures are a result of the ODR algorithm. We used the scipy.odr package. We added this information to the manuscript in Section 2.3, L203: "The regression was calculated with the Python scipy.odr package, which calculates the intercept as well as its uncertainty as standard deviation from the uncertainties of the input data."*

2. L224: authors mentioned that "the signature of all inflow samples...indicates that the CH$_4$ enhancements in the upwind boundary layer are mostly biogenic, but with a fossil influence". But from Figure 4, the IN symbol and its error bars are all located in green shaded area, i.e., EMID modern microbial. I do not fully understand why there is a fossil influence.
*True, the IN symbol and error bars are fully located in the green shaded area, but the signature is shifted towards the fossil fuel signatures compared to the FT (free troposphere) signature. If we assume that the free troposphere methane is from biogenic sources, then the inflow methane either is mainly biogenic with fossil influence or from a different type of biogenic sources. So, it could be that inflow sources are rather from waste management, which has more positive signatures, than agriculture, wetlands or ruminants, which are more negative. We added this to the text.*

3. L 250, Table 1: are the values of the isotopic composition and standard error originated from all flasks during each flight or only from the flasks in PBL? Please specify it.
I suppose the symbol "#" presents the number of flasks. Maybe No. is better for understanding.
*Only flasks in the PBL are considered here. We specified this and changed # to No.*

4. L253, Figure 4: I think the individual source signatures (with black symbols and numbers) represent the results originate from the flasks in PBL. If so, please specify this information, otherwise, it is misleading.

*We also specified this here.*

5. L267: "δ13C signatures from all samples are in the same range (not shown)". It would be interesting to see the results. Maybe put it in the appendix.
*We added Figure A1 in the appendix to show the Pniowek samples from inside and outside the ventilation shafts.*

[Figure]

*Figure A1: Comparison of coal mine signatures from samples taken outside and inside the ventilation shafts.*

6. L269: "...with one signature at -38 ‰." I assume that you mean this value is out of the main range. Perhaps change it to "with one biased signature at -38 ‰".
*Good idea. This is not exactly a biased signature though, but an outlier. We specified this.*

7. L278: "The aircraft samples showed average $\delta^2H$ signatures for the southern region, but the southeastern mines had lower $\delta^2H$ signatures than the entire USCB." I assume the aircraft samples indicate the dots in Figure 3, and the cyan square symbols in Figure 4. If so, in my opinion it would be beneficial to show another figure (like in the appendix) to better present the statement. The coordinate-related information cannot be found from Figure 3 or 4. If the "aircraft samples" represent the isotopic source signatures in PBL shown in Figure 4., it is the southwest but not the southeastern mines having lower $\delta^2H$ signatures. Please comment.
*Indeed, this refers to the isotopic source signatures in the PBL shown in Figure 4 and should name the southwestern mines with the lowest $\delta^2H$ signatures. We simplified and corrected the sentence to: "The $\delta^2H$ source signatures in the PBL derived from aircraft samples also showed that the southwestern region had the lowest $\delta^2H$ signatures (Figure 4)."*

8. L339: "For comparison, the EMID includes $\delta^2H$ signatures from 7 landfills (-275 ± 21 ‰) and signatures from 6 wastewater facilities (-323 ± 14 ‰) across Europe. The average signature over all these data points is -297 ± 30 ‰." I assume that the average signature from the 7+6=13 sites is calculated based on weighted average. How about the uncertainty? Is the error propagation or other method used? Please comment.
*The average was calculated from the 13 individual values. Also, the uncertainty gives the standard deviation of all 13 signatures. We agree that it makes more sense in this case to use a weighted average and error propagation, because these are two different source categories and the combination of their signatures would not increase the uncertainty range. We changed our method and the new values are 297 ± 18 ‰. We changed this in the text: "The weighted average of the signatures of the two sectors is -297 ± 18 ‰. The uncertainty was calculated through error propagation"*

9. L343: "we assume that the USCB $\delta^2H_{bio}$ signature for waste emissions is -300 ± 20 ‰ for our study". Is the mean value of -300‰ from the global modeling as mentioned in L341? This value does not match with the mean value of biogenic signatures shown in Figure 7. Additionally, from where the uncertainty of 20‰ come? Please specify.
*The average signature of -300‰ is a combination of the global modeling value and the EMID value. The 20‰ uncertainty is taken from the EMID observation uncertainty ranges. We added this to the manuscript.*

10. L351: CAMS-REG-GHG inventory has been updated to v5.3 in 2022. There might be no huge difference between v3.1 and v5.3. It is for your information.
It would be also beneficial to compare the spatial distribution of gridded inventory and the results here. Will the CAMS inventory in the southwest area tends to have more biogenic sources?

*We conferred with the creators of CAMS. The emission inventory v5.3 is only the global inventory. The regional inventory CAMS-REG-GHG has also been updated to a version 4.1, which includes the year 2018, when our measurements took place. Although the data is not publicly available yet, it was provided to us. There are no big changes.*
*That the southwest USCB has more biogenic sources is not a result or conclusion from our study. The lower signatures in $d^{13}C$ and $d^2H$ in the southwest are probably due to regional gradients in the coal mine methane signatures rather than sectorial fractionation changes because the gradient is observed both in the ground-based as well as the airborne data.*

**Technical comments:**
11. L100: the subscript in the "$\delta_2H$" should be superscript. – *Changed.*

12. L162: not fully understand the sentence "For each of these categories we determined the mean isotopic signature from all flights combined and for PL samples also for individual flights." Please rephrase.
*We rephrased to: For each of these categories we determined the mean isotopic signature for the entire campaign. Using the PL samples from each flight individually, we calculated the isotopic signatures of the individual target regions.*

13. L199: please keep consistent format for $\delta_x$ in equation (3) and afterwards. The $\delta_{obs}$, $\delta_{bg}$ are in bold in previous text.
*We changed all the bold face into normal font in the equations and text.*

14. L264: I think this sentence has a grammatical error. "This variability may result from different areas of the mine being exploited as longwalls at different depths of the mine are opened up or shut down during excavation."
Maybe change to "This variability may result from different areas of the mine during longwall exploitation at different depths of the mine which are opened up or shut down during excavation" Please rephrase.
*We changed this to: Within one mine the isotopic signatures differ due to the geographical structure. The signature of the ventilated methane then also varies with time, because longwalls at different depths of the mine are opened up or shut down during excavation.*

---

## Author Comment (AC2)

Answer to referee comment 2 for "**Source attribution of methane emissions from the Upper Silesian Coal Basin, Poland, using isotopic signatures**"

*We would like to thank the reviewer for the suggestions to improve the manuscript. Below you find our answers to their comments. The reviewer's comments are written in normal font, our answers in italics.*

The authors use airborne and ground-based air samples to attribute methane emissions in the region to sources, with a focus on distinguishing coal/fossil sources from biogenic/waste sources. The ground-based samples are used to identify source signatures and then, the airborne samples are used to determine the contribution of the two source types. Overall, the study is nicely written and presented but can benefit from some editing to clarify the analysis and results.

An interesting finding from this paper is the relatively large contribution of biogenic sources in the Upper Silesian Coal Basin. Given this finding, there is a need for some introduction of these biogenic sources in the Introduction and annotations in Figure 2. Although there is some text on this towards the end, additional text earlier on would be helpful.

It appears that a lot of the data being used in this paper have been published in previous papers. However, it's unclear if there's new data presented in this paper. This should be clarified (see detailed comments below).

*Thank you for the positive feedback and good suggestions. We improved the introduction of biogenic sources and clarified where data has been published previously. All airborne isotopic measurements are unpublished, but ground-based observations have been published previously with different focus.*

Below are detailed line-by-line comments:

L20: Replace "growth" with "concentration accumulation". – *We decided to use the term "increase of atmospheric methane levels".*

L20: Replace "on global" with "at global". – *Done.*

L24: Are the "other anthropogenic sources" related to the coal mine or something else? Landfills and wastewater should be listed here in the abstract given the findings of the paper. – *Done.*

L30: What are all the sources in the region? This speaks to an earlier comment about clarifying the sources being studied.
*In this sentence the focus is not on the individual sources or even what kind of sources there are. We want to convey that the airborne methane plume contained a well-mixed plume of methane containing all emissions from within the study region. We changed the phrase to: "…methane from the entire region…"*

L35: "allowed for the determination of the source signatures..."
*We changed this to: "We determined the source signatures of individual coal mine ventilation shafts using ground-based samples."*

L40: I think the main point is that the d$^2$H is important for source attribution and that the d$^2$H of the ventilation shafts differ from the regional d$^2$H values. Therefore, it would be worth rephrasing this sentence to highlight this point. The fact that the d$^2$H of ventilation shafts match previous studies is relevant but should be the secondary point.
*This is true and we changed the sentence accordingly omitting the comparison with previous studies. This secondary point is now only mentioned in the Discussion and Conclusions.*

L41-42, 34: Because wetlands and ruminants were mentioned in L34, I expected these to be significant. Suggest revising L34.
*We omitted the naming of source categories in line 34 to avoid confusion. Wetlands and ruminants are probably significant in the upwind and free tropospheric source signatures though.*

L45: It would be good to provide some quantitative comparison of underestimation. Either state that the common inventories estimate 6% biogenic or state the difference as a factor.
*We added the fractions given in the inventories (0.4-14%). This is also stated in the Conclusions.*

L50: "at limiting" to "to limit" – *Done.*

L52: The Global Methane Pledge is specifically for methane emission reductions, not all greenhouse gas emission reductions. Replace "greenhouse gas" with "methane". – *Done.*

L53: Replace "localize" with "locate". – *Done.*

L116: Define what is meant by "regional perspective". What is the scale of "regional"? It sounds as though the authors are re-analyzing existing data. Either way, this needs to be clarified. A clearer description of what new analysis is being performed here would be helpful.
*The region considered here is the USCB. We changed the paragraph to: In this study, we present isotopic methane sample analysis for the USCB. We analyze unpublished samples taken on a small aircraft and compare to already published ground samples to determine the contributions of coal mining and waste treatment to the total USCB methane emissions.*

L136: Pniowek is not shown in Figure 2. It should be identified. – *We marked it in the Figure.*

L141: Define FUB at first mention. – *We spelled out "Freie Universität Berlin", because it only appears once in the text.*

L153-154: Were the data published already? This needs to be clarified.
*The airborne data from the DLR Cessna Caravan has not been published previously. We added this information to the manuscript.*

L171: "...concentrations and isotopic compositions of CH$_4$ were..."
*This sentence seems correct to us.*

L218: how can the standard errors be small if the variance is large?

*What we meant here is that we have a large range in the concentration values and thus could constrain the source signature well. We rephrased to "large range of concentrations".*

L226: the location of Silesia needs to be shown in Figure 2. – *We marked it in the Figure.*

L261: the locations of the cow farm, landfills, manholes and wastewater facilities need to be shown in Figure 2. – *We marked the cow farm and landfill 2 in the Figure. The biogenic sources are not on the map, but 100 km to the east in Krakow.*

L266: what is meant by vicinity? In the conclusions, 1-2 km is mentioned. It's surprising that the samples taken directly within the shafts are similar to those taken 1-2 km away.
*We added the correct distance of 1-2 km also here in the text. The similarity of isotopic signatures from inside and outside the ventilation shafts seems logical. Although the concentrations in the samples were different, the enhancements outside the mines were large enough for a thorough Keeling analysis. The resulting source signatures are free of the influence of the background methane and independent of the sample concentrations. The signatures are not identical, though, because they were not sampled on the same day.*

Figure 5: Specify where in the mines the "mining" samples were taken. – *Done.*

Figure 5: Is the data from the MEMO2 dataset? Also specify that this is from ground-based studies.
*The ground-based data in our analysis is from the MEMO2 dataset, which in turn is part of EMID. We clarified this here.*

Figure 6: Specify whether the data is from ground-based measurements in the caption. – *Done.*

Figure 7: specify that EMID fossil fuel data is ground-based.
Overall, there are a lot of acronyms that are used interchangeably. For the ground-based data, it's referred to as MEMO2, EMID, and "ground-based". I suggest simply calling it "ground-based", if possible, and not using multiple names for the same datasets.
*We now only use two names for the ground data: "ground-based" are all samples in the USCB that were used in this study. "EMID" signatures and data applies to all samples taken all over Europe. "MEMO2" has been eliminated except for in the Acknowledgements.*

L324: Given the importance of the waste sector in this region, there is a need to describe these sources more. How big are the landfills and wastewater treatment plants both spatially and in terms of waste volumes? Also, is there an inventory of all methane sources in the USCB?

*There really is not a lot of information available on the waste sector in the region as there does not exist an emission database dedicated to the waste sector. However, there are some information where the landfills are located and how much waste is being deposited every year. Population density normally gives an idea of the amount of wastewater produced in a certain region. This information needs to be translated into methane emissions though. A rough estimate gives emissions in the order of 60-80 kt/year for the landfills. This will be topic of a planned future publication.*

*There are the Industrial Reporting (IR) emission database (ED) (former E-PRTR), the CoMet v4.01 ED (Gałkowski et al., 2021) and scientific gridded ED like EDGAR and CAMS-REG. Within CoMet v4.01 ED, 32 landfill locations are listed. Most of them are not reporting emissions of $CH_4$ to IR. We did not investigate this in depth, but in most cases, they are relatively small and their individual emissions fall below reporting threshold. Of those that did report emissions to E-PRTR, the reported numbers were around 3 kt/year for 8 to 14 reporting landfills. That is an order of magnitude smaller then what is estimated based on the amount of trash deposited. Additionally, 24 individual wastewater treatment plants are identified within CoMet v4.01 ED. None was reporting $CH_4$ in 2018. The numbers from gridded data like CAMS-REG should be consistent with E-PRTR.*

*We added some of this information to the manuscript.*

L333: the better discrimination of d$^2$H signatures than d$^{13}$C is interesting. Therefore, it would be helpful to see what the error would be if only d$^{13}$C was used in source attribution.
*The source attribution is not really possible with $\delta^{13}C$. The uncertainties of the signatures overlap considerably. Although we did calculate an average signature for the USCB coal mine emissions from the ground-based data, this value is very uncertain in $\delta^{13}C$ (-49.8 ± 5.7 ‰). The error using this value against the total USCB signature of -50.9 ± 0.7 ‰ would be around 500%.*

L359: can the locations of these landfills and wastewater treatment plants be shown in Figure 2?
*Krakow and the landfills and wastewater treatment plants are not in the area depicted by Figure 2, but about 50 km to the east of the eastern USCB border. We highlighted landfill 2 in the new Figure 2.*

Figure 8: The authors assume that waste is the dominant biogenic source in the area. However, there may be natural (and agricultural) sources that may be contributing more than assumed. The authors are probably right but it may be worth pointing out that there still are uncertainties in the biogenic methane source.
*A contribution from natural sources is possible. Especially agricultural sources may also be underestimated in the gridded inventories. This would influence our results because their isotopic methane signatures are at the lowest edge of the spectrum.*

L398: how much do the signatures vary over time? How does this affect uncertainties and the comparisons/analysis presented in this paper?
*Ground-based samples taken in 2018 and 2019 show no trend in the signatures from the coal mines (Figure 5). The $\delta^{13}C$ signatures of previous studies (Kotarba, 2001) have been lower than our values (Figure 7), but the $\delta^2H$ is in the same range. Since we only use $\delta^2H$ in the source apportionment we do not expect an influence of this shift. It is also hard to determine whether this is a real shift over time or if it is a result of the different techniques used.*

*We only have airborne signatures for the year 2018 and cannot deduce if the signature of the total emissions of the USCB changed over time.*

Gałkowski, M., Fiehn, A., Swolkien, J., Stanisavljevic, M., Korben, P., Menoud, M., Necki, J., Roiger, A., Röckmann, T., Gerbig, C., and Fix, A.: Emissions of CH4 and CO2 over the Upper Silesian Coal Basin (Poland) and its vicinity, ICOS ERIC - Carbon Portal, 10.18160/3K6Z-4H73, 2021.

Kotarba, M. J.: Composition and origin of coalbed gases in the Upper Silesian and Lublin basins, Poland, Organic Geochemistry, 32, 163-180, 10.1016/S0146-6380(00)00134-0, 2001.

---

## Author Comment (AC3)

Answer to referee comment 3 from Amy Townsend-Small for "**Source attribution of methane emissions from the Upper Silesian Coal Basin, Poland, using isotopic signatures**"

*We would like to thank Amy Townsend-Small for the suggestions to improve the manuscript. Below you find our answers to their comments. The reviewer's comments are written in normal font, our answers in italics.*

A couple of major takeaways: Some coal mines have methane formed through biogenic carbonate reduction, and some coal mines can emit thermogenic methane more isotopically similar to natural gas. You could highlight your results more clearly in the abstract – it seems that your study, using both isotopes, identifies these mines as clearly emitting thermogenic methane. I believe most papers have previously only used carbon isotopes? E.g. Zazzeri et al., 2016.

*We highlighted the differences between the coal mines and the advantage of using the hydrogen isotope more clearly in the abstract:*

*"Different layers of the USCB coal contain thermogenic methane, isotopically similar to natural gas, and methane formed through biogenic carbonate reduction. The signatures vary depending on what layer of coal is mined at the time of sampling."*

*"[The isotopic signature] clearly differs from the USCB regional signature in $\delta^2H$. This makes a source attribution using $\delta^2H$ signatures possible, which would not be possible with only the $\delta^{13}C$ isotopic signatures."*

Another thing that your study clearly shows is that the use of hydrogen stable isotopes is really essential for source apportionment in regions with a mix of thermogenic and biogenic sources. I was excited to see this result because I have also found similar results in Los Angeles, the Barnett Shale, and in Denver, Colorado. I see you mention this in the last sentence of your abstract – can you highlight it more? For example, I think Figure 5 also illustrates this well. In your Figure 5, for carbon isotopes, your mining samples are both above and below the isotopic composition of air. This makes it very, very difficult to use carbon isotopes as a tracer of this source in air. This has global implications because carbon isotopes are being used to track methane sources at background monitoring sites – not hydrogen! My group has found similar results with ground samples taken at natural gas methane sources – as an example see Townsend-Small et al., 2016, Geophysical Research Letters - Using stable isotopes of hydrogen to quantify biogenic and thermogenic atmospheric methane sources: A case study from the Colorado Front Range.

*Thanks for this feedback. We added the reference to the interesting Townsend-Small et al., (2016) paper.*

*We highlighted the importance of the hydrogen isotope signatures more clearly at the end of the abstract and conclusions.*

*"The average signature from the ventilation shafts of -49.8 ± 5.7 ‰ in $\delta^{13}C$ and -184 ± 32 ‰ in $\delta^2H$ clearly differs from the total regional signature in the $\delta^2H$ and makes a source*

*apportionment between coal mine and other emissions possible. This would not be possible with only the $\delta^{13}C$-CH$_4$ signatures, because the coal methane signatures vary considerably in $\delta^{13}C$ and are both above and below the isotopic composition of air."*

*"This study confirms the importance of $\delta^2H$-CH$_4$ observations for methane source apportionment, as reported in previous recent studies (Townsend-Small et al., 2016; Fernandez et al., 2022). This is especially true in regions with a mix of thermogenic and biogenic sources and large variations in the $\delta^{13}C$ signature of one sector."*

Fernandez, J. M., Maazallahi, H., France, J. L., Menoud, M., Corbu, M., Ardelean, M., Calcan, A., Townsend-Small, A., van der Veen, C., Fisher, R. E., Lowry, D., Nisbet, E. G., and Röckmann, T.: Street-level methane emissions of Bucharest, Romania and the dominance of urban wastewater, Atmospheric Environment: X, 13, 100153, 10.1016/j.aeaoa.2022.100153, 2022.

Townsend-Small, A., Botner, E. C., Jimenez, K. L., Schroeder, J. R., Blake, N. J., Meinardi, S., Blake, D. R., Sive, B. C., Bon, D., Crawford, J. H., Pfister, G., and Flocke, F. M.: Using stable isotopes of hydrogen to quantify biogenic and thermogenic atmospheric methane sources: A case study from the Colorado Front Range, Geophysical Research Letters, 43, 11,462-411,471, 10.1002/2016GL071438, 2016.